# F18:A-:B1 Plasmids Carrying *bla*_CTX-M-55_ Are Prevalent among *Escherichia coli* Isolated from Duck–Fish Polyculture Farms

**DOI:** 10.3390/antibiotics12060961

**Published:** 2023-05-25

**Authors:** Li-Juan Zhang, Jin-Tao Yang, Hai-Xin Chen, Wen-Zi Liu, Yi-Li Ding, Rui-Ai Chen, Rong-Min Zhang, Hong-Xia Jiang

**Affiliations:** 1Zhaoqing Branch Center of Guangdong Laboratory for Lingnan Modern Agricultural Science and Technology, Zhaoqing 526000, China; 2Guangdong Key Laboratory for Veterinary Pharmaceutics Development and Safety Evaluation, College of Veterinary Medicine, South China Agricultural University, Guangzhou 510642, China; 3Life Science Department, Foshan University, Foshan 528000, China; yding@kean.edu; 4Guangdong Laboratory for Lingnan Modern Agriculture, Guangzhou 510642, China

**Keywords:** duck–fish polyculture farm, *Escherichia coli*, antibiotic resistance, *bla*
_CTX-M-55_, F18:A-:B1, transmission

## Abstract

We determined the prevalence and molecular characteristics of *bla*_CTX-M-55_-positive *Escherichia coli* (*E. coli*) isolated from duck–fish polyculture farms in Guangzhou, China. A total of 914 *E. coli* strains were isolated from 2008 duck and environmental samples (water, soil and plants) collected from four duck fish polyculture farms between 2017 and 2019. Among them, 196 strains were CTX-M-1G-positive strains by PCR, and 177 (90%) *bla*_CTX-M-1G_-producing strains were *bla*_CTX-M-55_-positive. MIC results showed that the 177 *bla*_CTX-M-55_-positive strains were highly resistant to ciprofloxacin, ceftiofur and florfenicol, with antibiotic resistance rates above 95%. Among the 177 strains, 37 strains carrying the F18:A-:B1 plasmid and 10 strains carrying the F33:A-:B- plasmid were selected for further study. Pulse field gel electrophoresis (PFGE) combined with S1-PFGE, Southern hybridization and whole-genome sequencing (WGS) analysis showed that both horizontal transfer and clonal spread contributed to dissemination of the *bla*_CTX-M-55_ gene among the *E. coli*. *bla*_CTX-M-55_ was located on different F18:A-:B1 plasmids with sizes between ~76 and ~173 kb. In addition, the presence of *bla*_CTX-M-55_ with other resistance genes (e.g., *tetA*, *floR*, *fosA3*, *bla*_TEM_, *aadA5 CmlA* and *InuF*) on the same F18:A-:B1 plasmid may result in co-selection of resistance determinants and accelerate the dissemination of *bla*_CTX-M-55_ in *E. coli*. In summary, the F18:A-:B1 plasmid may play an important role in the transmission of *bla*_CTX-M-55_ in *E. coli*, and the continuous monitoring of the prevalence and transmission mechanism of *bla*_CTX-M-55_ in duck–fish polyculture farms remains important.

## 1. Introduction

Antimicrobial resistance is a serious global public health problem associated with significant clinical, economic and social impacts. *Escherichia coli* exists as part of the commensal microbiota in the mammalian digestive tract, as a zoonotic pathogen responsible for intestinal and extraintestinal infections in both humans and animals [1,2]. Globally, the emergence of multidrug-resistant (MDR) *E. coli* producing extended-spectrum β-lactamase (ESBL) enzymes has led to empirical therapy failure, leading to high morbidity and mortality, which has raised great public concern [1,2].

Currently, CTX-M-bearing *E. coli* is the most common species related to ESBLs and more than 220 CTX-M family members have been identified. These variants were divided into five major groups (groups 1, 2, 8, 9 and 25) based on their amino acid homology. Among these groups, 1 and 9 were the most common globally [2,3,4]. Over the past decade, CTX-M-55, as a variant of CTX-M-15, from animal-origin *E. coli*, which was first discovered in Thailand in 2006, has spread rapidly in dozens of countries around the world, especially in China [4,5,6]. The dissemination of *bla*_CTX-M-55_ was mainly caused by plasmid-mediated gene horizontal transfer, and epidemic self-mobilizable F33:A-:B, IncI1, IncI2 and IncHI2 plasmids played an important role in the transmission of *bla*_CTX-M-55_ [6,7,8]. In addition to animals, *bla*_CTX-M-55_ is also distributed in food products and humans, and frequently co-localized with other resistance genes, such as *fosA3*, *rmtB*, *mcr-1*, *bla_TEM_*, *tet(A)* and *floR* [7,8,9,10]. The wide distribution of *bla*_CTX-M-55_ and the co-transfer of *bla*_CTX-M-55_ with different resistance genes worldwide represent a growing threat to public health.

As the largest producer and consumer of cultivated duck in the world, duck production plays a major role in the agricultural economy of China [11]. Duck farming in China is practiced on a large and diverse scale, and the integrated culture of fish–duck farming using untreated duck manure as fish feed, is the typical farming method throughout coastal areas of China, particularly in Guangdong Province. Previous studies have shown that fish–duck integrated farming systems have become a hotspot environment for the occurrence and proliferation of antibiotic resistance genes (ARGs) and antibiotic-resistant bacteria (ARB), which can facilitate the spread of resistance genes and have the potential to be the reservoir of novel ARGs [12,13,14]. Owing to the threat of *bla*_CTX-M-55_ to public health and the risks of ARG transmission in fish–duck farming, monitoring the prevalence of *bla*_CTX-M-55_-positive *E. coli* isolates from duck–fish polyculture farms should receive more attention.

However, CTX-M-55 has been widely reported in *E. coli* isolated from food animals, pets and humans in China [6,8,15]. Current data on the prevalence, genetic information and the transmission mechanism of CTX-M-55-producing *E. coli* from duck–fish polyculture farms are still limited. In this study, we investigated the prevalence of *bla*_CTX-M-55_ and illustrated the characteristics of *bla*_CTX-M-55_-bearing *E. coil* and plasmids recovered from ducks and the environment in duck–fish polyculture farms. Our findings emphasize the importance of the surveillance of *ESBL*-producing *E. coli* in duck–fish polyculture farms in China and provide knowledge for further One Health studies to control the spread of resistant bacteria from food animals to humans.

## 2. Results

### 2.1. Identification of bla_CTX-M-55_-Positive E. coli Isolates

A total of 196 *bla*_CTX-M-1G_-positive *E. coli* isolates were obtained from 2008 different samples, and the *bla*_CTX-M-1G_-producing *E. coli*-positive rates from duck fecal samples, water, soil and foliage were 14.4% (138/957), 6.25% (31/496), 1.5% (7/476) and 1.3% (1/79), respectively (Appendix A). Six different CTX-M-1G subtypes were obtained from the 196 *bla*_CTX-M-1G_-positive strains, including CTX-M-55 (177 strains), CTX-M-79 (8 strains), CTX-M-123 (5 strains), CTX-M-3 (3 strains), CTX-M-64 (2 strains) and CTX-M-224 (only 1 strain). Among the 177 CTX-M-55-positive strains, 77.9% (138/177) were isolated from fecal samples, 17.5% (31/177) were isolated from water, 4.0% (7/177) were isolated from soil and only 1 (1/177, 0.6%) isolate was obtained from foliage (Appendix A).

The antimicrobial susceptibility results showed that the 177 *E. coli* isolates were highly resistant to ceftiofur, ciprofloxacin and florfenicol, with resistance rates of 98.3%, 96.7% and 95.4%, respectively. The rates of resistance to colistin, ceftazidime and fosfomycin were 35.0%, 31.7% and 29.9%, respectively. However, these strains were less resistant to tigecycline, meropenem and amikacin, and the resistance rates were less than 10% (Appendix A).

To further understand the molecular characteristics of *bla*_CTX-M-55_-positive *E. coli*, the PCR-base replicon typing (PBRT) method was performed on the 177 strains. The results showed that 19.2% (34/177), 6.2% (11/177), 1.7% (3/177) and 1.1% (2/177) of the strains carried the IncT, IncN, IncHI1 and IncP plasmids, respectively, and 10.7% (19/177) and 5.6% (10/177) of the strains carried IncY and IncI1 plasmids, respectively. Notably, 81.4% (144/177) of the strains carried IncF plasmids. We further performed F plasmid replicon sequencing typing (RST) on these 144 IncF-type plasmid-carrying strains. The results showed that F18:A-:B1-plasmid-carrying strains accounted for the largest proportion, with 37 (37/144, 25.7%) strains, followed by F33:A-:B-plasmid-carrying strains, with 10 (10/144, 6.9%) strains carrying this type of plasmid and the remaining 97 strains carrying other types of IncF plasmids.

Based on the above results, we selected the 37 F18:A-:B1-plasmid-carrying strains and 10 F33:A-:B-plasmid-carrying strains for further study of the characteristics and transmission mechanism of plasmids carrying *bla*_CTX-M-55_-positive *E. coli*. Among the 47 isolates, the number of strains from duck fecal samples was the largest, accounting for 81% (38/47), followed by the strains from water samples, accounting for 13% (6/47). The remaining 6.4% (3/47) of strains were isolated from soil samples (Figure 1).

### 2.2. Molecular Characterization of bla_CTX-M-55_-Positive E. coli

PFGE was successfully performed for all 47 selected strains, and the PFGE results were divided into 15 different clusters according to similarity >85%, indicating the genetically different backgrounds of strains from different sources. In addition, two clusters of strains with the same PFGE spectrum were derived from the different types of samples from farms B, C and D (TC5, UD5, RD1, B1W1, UD9 and D7), suggesting that there is clonal transmission among different farms (Figure 1).

The conjugation results indicated that 17 of the 47 strains could successfully transfer *bla*_CTX-M-55_ from the donor strain to the recipient strain *E. coli* C600, and the transfer frequency was between 2.6 × 10^−5^ and 4.65 × 10^−1^ (Appendix A). There were four strains that could transfer *bla*_CTX-M-55_ from the donor strain to the recipient strain *E. coli* J53 with conjugative transfer frequencies between 3.89 × 10^−6^ and 8.35 × 10^−5^. The remaining 26 strains showed unsuccessful transfer of *bla*_CTX-M-55_ from the donor strain to the recipient strain after multiple attempts (Appendix A). To further determine the location of the *bla*_CTX-M-55_ gene in the non-conjugatively transferable strains and the genetic environment of the *bla*_CTX-M-55_ gene, whole-genome sequencing was performed on all 47 strains. Sequence analysis showed that among the 26 non-conjugable strains, the *bla*_CTX-M-55_ gene was located on the chromosome of 11 strains, while the *bla*_CTX-M-55_ gene of the remaining 15 strains was located on plasmids.

### 2.3. Genomic Analysis of bla_CTX-M-55_-Positive E. coli

Whole-genome sequencing data were generated for the 47 *bla*_CTX-M-55_-positive *E. coli* isolates. The results of WGS demonstrated that these isolates belonged to seventeen distinct strains (STs). Among them, the most dominant ST type was ST602, with a total of nine (9/47, 19%) strains belonging to this ST type; followed by ST155, ST410 and ST2179 (five, 10.6% each); ST48, ST162 and ST354 (three, 6.4% each); and ST165 and ST224 (two, 4.3% each). Of the remaining eight ST types, only one strain belonged to each ST type.

We identified 14 ARGs that mediated resistance to 9 types of antibiotics that coexisted with *bla*_CTX-M-55_ in these 47 *E. coli* isolates. These included genes that mediated resistance to fosfomycin, colistin, tetracycline, aminoglycosides, chloramphenicol, quinolones, macrolides, sulfonamides and lincomycin. Among these, 26% (12/47) and 13% (6/47) of the strains carried the quinolone resistance genes *oqxAB* and *qnrS*, respectively. In addition, 38% (18/47) of the strains carried the colistin resistance gene *mcr-1*, and 26% (12/47) and 21% (10/47) of the strains carried the fosfomycin resistance genes *fosA3* and *fosA7.5*, respectively. Eleven percent (5/47) of the strains carried the chloramphenicol resistance gene *floR*, and thirteen percent (6/47) of the strains carried the lincomycin resistance gene *lnuF*. Interestingly, our results showed that all strains belonging to ST602 carried the fosfomycin resistance gene *fosA7.5*, and *fosA7.5* was located on the chromosome. Additionally, all ST410 strains carried the *lnuF* gene, and the *bla*_CTX-M-55_ gene carried by the ST410 strains in the current study could not be transferred by conjugation (Figure 1 and Appendix A).

Genetic environment analysis showed that *bla*_CTX-M-55_ was present in four genomic contexts, including types I, II, III and IV. The structure of type I was IS*Ecp1*-*bla*_CTX-M-55_-*orf477*, which was the most prevalent, and was isolated in a total of 21 (45%) strains originating from feces, soil and water (Appendix A and Appendix A). The structure of type II was ΔIS*26*-ΔIS*Ecp1*-*bla*_CTX-M-55_-*orf477*-ΔTn*2*, and there were 12 (26%) strains belonging to this genetic environment. There were 10 (21%) strains belonging to the type III genetic environment, and the structure of type III was ΔIS*26*-ΔIS*Ecp1*-*bla*_CTX-M-55_-*orf477*-ΔTn*2*-ΔIS*26*. The type IV genetic environment was more complex, and its structure was ΔTn*2*-IS*Ecp1*-*bla*_CTX-M-55_-*orf477*-ΔTn*2*-ΔIS*26*. Only three isolates belonged to this genetic environment. Notably, among the strains belonging to the type I genetic structure, the *bla*_CTX-M-55_ gene of the ten strains was located on the plasmid, and the *bla*_CTX-M-55_ gene of the other eleven strains was located on the chromosome. The *bla*_CTX-M-55_ gene in the strains belonging to the type II and type III genetic structures were all located on the plasmid. Among the strains with type IV genetic structure, one strain had the *bla*_CTX-M-55_ gene located on the chromosome, and *bla*_CTX-M-55_ in the other two strains was located on plasmids (Appendix A and Appendix A).

### 2.4. Complete Sequence Analysis of bla_CTX-M-55_-Carrying F18:A-:B1 Plasmids

To better understand the characteristics of the *bla*_CTX-M-55_-bearing F18:A-:B1 plasmid, S1-PFGE and Southern blotting were performed on the 12 conjugants successfully obtained from the F18:A-:B1 plasmid harboring *bla*_CTX-M-55_-positive strains. The results showed that *bla*_CTX-M-55_ from 12 strains was located on F18:A-:B1 plasmids with sizes between ~76 kb and ~173 kb (Appendix A). To further explore the sequence features of the F18:A-:B1 plasmid, four representative strains (PBS4, B1W1, B1S11 and KW21) were subjected to long-read sequencing to obtain the complete sequence of different F18:A-:B1 plasmids. The complete sequences of the F18:A-:B1 plasmids pPBS4, pB1W1, pB1S11 and pKW21 were obtained by long-read combined with short-read sequencing, and detailed information on the four plasmids is shown in Appendix A.

Sequence analysis revealed that all four plasmids were novel IncF18:A-:B1 plasmids, three (pBS4, pB1W1 and pKW21) of which carried *bla*_CTX-M-55_, and the remaining plasmid (pB1S11) did not carry *bla*_CTX-M-55_. In strain B1S11, *bla*_CTX-M-55_ was located on an IncHI2 plasmid. Linear comparison revealed that the backbone of the four F18:A-:B1 plasmids was similar to that of other typical IncF plasmids containing regions for functions of replication (*repA*, *repB*), conjugal transfer (*tra*, *trb*), partitioning (*parA*, *parB*) and maintenance and stability (*pemI*, *pemK*, *stbA*). Notably, in the plasmid pB1W1, ~29-kb conjugative regions containing the *traV*, *traR*, *traC*, *traW*, *traU*, *traN*, *traF*, *traH*, *traG*, *traD*, *traT*, *traS*, *trbD*, *trbG*, *trbI*, *trbB*, *trbJ* and *trbF* genes were lacking, which may explain the conjugative transfer failure of pB1W1 (Figure 2). Basic Local Alignment Search Tool (BLAST) analysis indicated that the four plasmids (pBS4, pB1W1, pB1S11 and pKW21) had high similarity to the *bla*_CTX-M-55_-carrying plasmid pTREC8 of an *E. coli* strain isolated from wetland sediment in the United States (GenBank accession no. MN158991.1) and shared high identities of 99.9%, 100%, 99.9% and 99.8%, respectively (Figure 2).

Although the F18:A-:B1 plasmid possessed a conserved backbone region, the variable regions, especially multidrug resistance regions, were distinct. The multidrug resistance region of pKW21 was 80,273 bp, which is quite different from the resistance regions of the other three F18:A-:B1 plasmids in this study. The ~80.3 kb MDR region contained 11 ARGs (*bla*_CTX-M-55_, *aac(3′)-IIa*, *aph(3′)-Ib*, *mph(A)*, *aph(6′)-Id*, *tet(R)*, *tet(A)*, *floR*, *bla*_TEM-1B_, *fosA3* and *aadA5*) with one or two copies, interspersed with different complete or truncated insertion sequences and transposons (IS*26*, IS*4* family, ΔTn*3*, ΔTn*2*, ΔIS*Ecp1*, ΔIS*1R*, IS*5075*, IS*91* and IS*Cfr1*) (Figure 3a). BLAST analysis showed that the ~80.3 kb region shared a high identity (>99.99%) with plasmids collected from humans and pigs that did not bear *bla*_CTX-M-55_, such as *E. coli* plasmid p14406-FII (GenBank accession no. MN823988), p13P484A-2 (GenBank accession no. CP019282) and pTEM-1-GZC065 (GenBank accession no. CP048026). Compared with p14406-FII, p13P484A-2 and pTEM-1-GZC065, the difference was that three regions of pKW21 (IS*26*-*aph(3′)-Ib*-IS*26*-*mph(A)*-IS*26*, IS*26*-*hp-hp-hp*-IS*26*-*hp-hp*-IS*26* and IS*26*-*hp*-*hp*-*hp*-ΔTn*2*-*hp*-*tet(R)-tet(A)*-*hp*-Δ*Tn2*-IS*26*-*hp*-*hp*-IS*26*) were inverted. Notably, there was a region of pKW21 (~33 kb) containing *bla*_CTX-M-55_ that completely mismatched with the compared plasmids (Figure 3a).

The MDR regions of pB1W1, pB1S11 and pPBS4 were ~46.0 kb, ~26.5 kb and ~38.9 kb, respectively. Even though the resistance genes in the three plasmids were varied, they all contained *floR*, *tet(A)* and *tet(R)* and *bla*_CTX-M-55_, except pB1S11. In addition to the mentioned genes, the pB1S1 MDR region also contained the quinolone resistance gene *qnrS1*, which was highly similar to pGSH8M-2-1 (GenBank accession no. AP019676.1) and shared 99.8% identify with 100% coverage. pGSH8M-2-1 was recovered from the effluent of a wastewater treatment plant in Tokyo Bay, and the only difference in the MDR region between pGSH8M-2-1 and pB1S11 was that an IS*26* was inserted between two ΔTn*2* in pB1S11 (Figure 3b). Compared with pB1S11, pB1W1 and pPBS4 contained more resistance genes and carried multiple copies of IS*26* and transposons. BLAST analysis did not detect plasmids with high homology to the MDR region of pPBS4 and pB1W1, which may suggest that multiple copies of IS*26* and transposons formed the distinctive MDR regions of pPBS4 and pB1W1 through multiple complex recombination events. Notably, although pPBS4 and pB1W1 were derived from the same farm (Farm B), the genetic environments of *bla*_CTX-M-55_ in pPBS4 and pB1W1 were different, indicating that the *bla*_CTX-M-55_ in these two plasmids originated from different sources (Figure 3b). In addition, we also found a truncated class I integron in the plasmid pB1W1 (Figure 3b).

## 3. Materials and Methods

### 3.1. Sampling Information, Bacterial Isolation and Identification

From March 2017 to August 2019, a total of 2008 nonduplicate samples including 496 water samples, 476 soil samples, 957 duck fecal samples and 79 foliage samples were collected from four duck–fish polyculture farms in the Panyu District of Guangzhou, China (Figure 1; see also Appendix A in the Appendix A). All samples were screened for cefotaxime-resistant *E. coli* by a selective isolation procedure. In brief, each sample was suspended in 10 mL of buffered peptone water (BPW; BD Difco, Sparks, MD, USA) and incubated at 37 °C for 24 h. Then, subsequent selective cultivation on MacConkey (MC; BD Difco) agar supplemented with 1 mg/L cefotaxime (CTX) was performed. For each sample, only one red colony was selected and identified as *E. coli* by MALDI-TOF MS AximaTM (Shimadzu-Biotech Corp., Kyoto, Japan) and 16S rRNA sequencing by Sanger sequencing. In all cefotaxime-resistant *E.coli*, *bla*_CTX-M-1G_ was detected by PCR using previously reported primers [16] and sequencing (Appendix A).

### 3.2. Antimicrobial Susceptibility Testing

Antibiotic susceptibility testing was performed by the agar dilution method and interpreted according to the Clinical and Laboratory Standards Institute guidelines (CLSI M100-S29) for the following antimicrobials: amikacin, meropenem, cefotaxime, ceftazidime, ceftiofur, florfenicol, ciprofloxacin and fosfomycin [17]. Susceptibility to colistin and tigecycline was assessed by broth microdilution as recommended by the European Committee on Antimicrobial Susceptibility Testing (EUCAST Version 9.0) [18]. *E. coli* ATCC 25922 was used as the quality control strain.

### 3.3. Molecular Typing

The incompatibility (Inc) groups of all *bla*_CTX-M-55_-producing *E. coli* were assigned by PCR-base replicon typing (PBRT) [19]. To better characterize IncFII plasmid, replicon sequencing typing (RST) was performed, according to protocols described previously [20]. Based on the results of PBRT analysis, a total of 47 strains (37 strains harboring the F18:A-:B1 plasmid and 10 strains harboring the F33:A-:B- plasmid) were selected for further study to explore the transmission characteristics and molecular mechanism of *bla*_CTX-M-55_ in these strains.

The genetic typing of the 47 selected *bla*_CTX-M-55_-producing *E. coli* isolates was performed by digestion with restriction endonuclease *XbalI* and pulsed-field gel electrophoresis (PFGE) according to our previous study [21]. The band patterns were analyzed with BioNumerics software version 5.10 (Applied Maths, Austin, TX, USA).

### 3.4. Conjugation Assay and Southern Blotting

To investigate the transferability of the resistance genes, a conjugation assay was performed for all selected *bla*_CTX-M-55_-positive *E.coli* isolates with streptomycin-resistant *E. coli* C600 or sodium-azide-resistant *E. coli* J53 as the recipient strain. For *E. coli* C600, donor strains and *E. coli* C600 were mixed and applied to a 0.22 μm filter in Luria-Bertani (LB) plates for 16~18 h. The mixed culture was then diluted and spread on selective MacConkey agar plates containing both 1 mg/L of cefotaxime and 2 g/L of streptomycin to recover transconjugants. For *E. coli* J53, donor strains and *E. coli* J53 were mixed and applied to a 0.22 μm filter in LB plates for 16–18 h. The mixed culture was then diluted and spread on selective MacConkey agar plates supplemented with 0.5 mg/L cefotaxime and 0.2 g/L sodium azide. Transconjugants were confirmed by PCR. S1-PFGE and Southern blotting were performed to determine plasmid size according to previous study, and the *Salmonella enterica* serotype, Braenderup H9812, was used as the standard size marker [22].

### 3.5. DNA Extraction and Whole-Genome Sequencing

Total DNA was extracted from 47 *bla*_CTX-M-55_-producing *E. coli* isolates using a Genomic DNA Purification Kit (TIANGEN, Beijing, China) according to the manufacturer’s instructions. WGS was performed with the Illumina HiSeq 2500 System (Novogene Guangzhou, China) using the paired-end 2 × 150-bp sequencing protocol. The draft genome was assembled using the tools available at EnteroBase (https://enterobase.warwick.ac.uk/species/ecoli, accessed on 28 November 2020) with default parameters. All genome assemblies of the 47 sequenced *E. coli* isolates were deposited in GenBank and are registered with BioProject number PRJNA934699. Then, the sequence types, replicon types and antibiotic resistance genes of all the sequenced isolates were identified by the Center for Genomic Epidemiology (http://www.genomicepidemiology.org/, accessed on 20 March 2021). Four representative strains (PBS4, B1W1, B1S11 and KW21) were further selected for whole-genome sequencing on the PacBio RS II sequencing platform (Biochip Company, Tianjin, China) to obtain the complete sequence of the F18:A-:B1 plasmid. Sequences of those strains were assembled using HGAP version 4.0. to analyze the genetic features. BRIG (Vision 0.95) and Easyfig (Vision 2.2.5) software were used for comparative analysis with other plasmid sequences published in NCBI. The complete sequence of the plasmid was annotated and analyzed using RAST (https://rast.nmpdr.org/rast.cgi, accessed on 12 March 2021), ISfinder (https://www-is.biotoul.fr/, accessed on 20 March 2021), Resfinder (https://cge.cbs.dtu.dk//services/ResFinder/, accessed on 22 March 2021) and BLAST (https://blast.ncbi.nlm.nih.gov/Blast.cgi, accessed on 23 March 2021).

### 3.6. Nucleotide Sequence Accession Numbers

The complete sequences of the plasmids (pPBS4, pB1W1, pB1S11 and pKW21) have been deposited in GenBank under accession numbers CP117716, CP117722, CP117718 and CP117673.

## 4. Discussion

Duck–fish polyculture is a common circular farming model in the Pearl River Delta in southern China, specifically in Guangdong Province. In this model, duck manure is discharged directly without treatment, and a large number of ARGs or residual agents can directly contaminate fish ponds, promoting the transmission of ARGs between ducks and fish [14,23,24]. Previous studies have shown that fish–duck integrated farming systems have become a hotspot environment for the occurrence and proliferation of ARGs and ARB, and both ARGs and pathogen-related ARB have been detected in the water and sediment of this culture system [12,13,14]. Moreover, fish–duck integrated aquaculture farms have significantly higher levels of antibiotic resistance compared to monoculture fish farms, suggesting a higher risk of transmission of ARGs and mobile genetic elements (MGEs) to humans or the environment [13]. The Pearl River Delta water system is intricate, which provides a unique opportunity to develop freshwater aquaculture, but there is also the risk of contaminating river or sea areas on a large scale by ARG and ARB dissemination via aquatic water or sediment [24,25]. Based on the “one health” concept, considering the common duck–fish freshwater aquaculture system in the Pearl River Delta in southern China, greater attention should be given to the transfer risk of ARGs in integrated duck–fish farming to promote the healthy development of Chinese aquaculture and the environment.

In this study, the antibiotic susceptibility of 177 *bla*_CTX-M-55_-positive strains was tested with 10 antibiotics commonly used in both veterinary and human medicine. The results showed that the *bla*_CTX-M-55_-positive strains were highly resistant strains, and almost all strains were multidrug-resistant. In addition to resistance to cephalosporins, they were also highly resistant to ciprofloxacin and florfenicol (with resistance rates at 96.7 and 95.4%, respectively). These strains also had a high resistance rate to ceftazidime (31.7%), colistin (35.0%) and fosfomycin (29.9%), and the resistance rate to tigecycline, meropenem and amikacin was less than 10%. Two recent studies also showed that *E. coli* strains isolated from ducks and the environment were not only resistant to cephalosporins but also resistant to chloramphenicols, aminoglycosides, quinolones and tetracyclines. Our results were similar to those reported in their study. Notably, the rate of detection of strains carrying the *bla*_CTX-M-55_ gene was higher than that of strains without this gene for both the antibiotic resistance spectra and ARGs [10,26].

Previous studies have shown that *bla*_CTX-M-55_ is mainly transmitted by the horizontal transfer of epidemical plasmids, and IncI1 and F33:A-:B- plasmids were the most important types of plasmids that mediated the spread of *bla*_CTX-M-55_ in human and animal *E. coli* in China [6,15]. In the present study, we selected 37 *bla*_CTX-M-55_-positive strains carrying the F18:A-:B1 plasmid and 10 *bla*_CTX-M-55_-positive strains carrying the F33:A-:B- plasmid to explore the transmission mechanism and molecular characteristics of *bla*_CTX-M-55_ among these strains. The PFGE results showed that the 47 strains could be divided into 15 different clusters, and the genetic backgrounds were relatively different. However, strains from different farms also had the same PFGE profile. CTX-M-positive *E. coli* isolates from ducks in Korea also had significant differences in PFGE profiles, but the same PFGE profiles were found in different livestock farms and slaughterhouses, which was consistent with the results of our study [27].

Multilocus sequence typing results indicated that ST602, ST155, ST410 and ST2179 were the prevalent ST types in our study, and the internationally prevalent ST10 and ST131 clones were not detected. ST602 has not been widely reported in previous studies, but a recent epidemiological surveillance of ESBL *E. coli* from human and food-chain-derived samples from England, Wales and Scotland found that the CTX-M-1G-positive ST602 strain was widely present in chicken samples [28]. Notably, all ST602 strains carried the *bla*_CTX-M-55_ gene chromosomally, and co-carried the chromosomal fosfomycin resistance gene *fosA7.5*. *fosA7.5* is a new member of the fosfomycin resistance gene *fosA7* gene family recently reported in *E. coli* isolates in Canadian hospitals. Its distribution is limited to *E. coli*, and it can be located on both plasmids and chromosomes [29]. Because fosfomycin was effective as a first-line therapy for urinary tract infections, the emergence of f*osA7.5* and *bla*_CTX-M-55_ co-carrying ST602 clone needs more attention.

ST410 *E. coli* is an emerging multidrug-resistant pathogen. Two major sublineages are currently circulating in Europe and North America, one is a fluoroquinolone- and extended-spectrum cephalosporin-resistant clade that emerged in the 1980s, and the other is a carbapenem-resistant clone that emerged in 2003. ST410 has been considered a “high-risk” clone similar to ST131 owing to its high transmissibility, its capacity to cause recurrent infections and its ability to persist in the gut [30,31]. This clone has also been found to harbor *mcr-1* in isolates recovered from food and human samples worldwide, and *tet(X)*-carrying ST410 *E. coli* in China and South Asia has been recently reported [10,31,32,33]. Given the potential of ST 410 *E. coli* to acquire resistance to last-resort antimicrobials, this clone should arouse regional and global concern.

The genetic contexts of *bla*_CTX-M-55_ were divided into four types in the current study. In the genetic context of type III (ΔIS*26*-ΔIS*Ecp1*-*bla*_CTX-M-55_-*orf477*-ΔTn*2*-ΔIS*26*) and type IV (ΔTn*2*-IS*Ecp1*-*bla*_CTX-M-55_-*orf477*-ΔTn*2*-ΔIS*26*), ΔTn*2* and ΔIS*26* were located both downstream and upstream of ΔIS*Ecp1*/IS*Ecp1*-*bla*_CTX-M-55_-*orf477*’s structure. This structure was found not only in *E. coli* but also in *Klebsiella*, *Vibrio parahaemolyticus* and *Salmonella*. This structure, possible formed by a copy of IS*26* and an *ISEcp1*-mediated transposon carrying *bla*_CTX-M-55_ and *orf477*, was inserted into a *tnpA* gene; this finding stresses the need for further assessment of the mobility of IS*26* or its variant [34].

Plasmids play a key role in the horizontal transfer of the ESBL gene among *E. coli* strains. Our results suggest that the F18:A-:B1 plasmid may play an important role in the transmission of the *bla*_CTX-M-55_ gene in *E. coli* isolated from duck farms and the environment. The F18:A-:B1 plasmid was first reported in avian pathogenic *E. coli* in 2010. It is characterized by the lack of an iron uptake gene (*eitABCD*), hemagglutinin and a survival gene. The *bla*_CTX-M-55_-bearing F18:A-:B1 plasmid was first reported in *E. coli* isolated from patients with urinary tract infections in the United States in 2016, and carried the *mcr* gene [20,33]. Subsequently, the F18:A-:B1-plasmid-carrying *bla*_CTX-M-55_ was reported in *E. coli* isolated from human clinical samples in China [35]. Recently, a study in Southeast Asia reported that the highly pathogenic clone *E. coli* ST410 isolated from the environment and humans carried the *bla*_CTX-M-55_-bearing F18:A-:B1 plasmid with a high prevalence [30]. In addition, studies from Tunisia and China also reported the high prevalence of the F18:A-:B1 subtype in animal-derived *E. coli* strains, which also contained *fosA3*, *oqxAB*, *bla*_CTX-M-14_ and other resistance genes [36,37,38]. These studies emphasize the possibility of horizontal transfer of the F18:A-:B1 plasmid between humans and animals.

However, to the best of our knowledge, only one recent study reported *bla*_CTX-M-55_, *floR* and *fosA3* carrying the F18:A-:B1 plasmid obtained from ducks [10]. Our study is the first to report the high prevalence of the *bla*_CTX-M-55_-carrying F18:A-:B1 plasmid in *E. coli* isolated from duck–fish polyculture farms. Complete plasmid sequence analysis showed that *bla*_CTX-M-55_ was colocalized with *tetA*, *floR*, *fosA3*, *bla*_TEM_, *aadA5*, *aph*(3′)-*Ib*, *aph(6′)-Id*, *CmlA*, *InuF* and other ARGs on the F18:A-:B1 plasmid. These ARGs, linked by different kinds of insertion sequences and transposable sequences and multiple copies of IS*26*, constitute the complex multidrug resistance region of the F18:A-:B1 plasmid. IS*26* is commonly associated with ARGs in multidrug-resistant Gram-negative and Gram-positive species, and is most widespread in Gram-negative bacteria. Clusters of ARGs can be generated by directly oriented IS*26* interspersed in multiple resistant pathogens [34,39]. As the best-studied member of the IS*26* family, IS*26* is known to form cointegrates using conservative transposition, homologous recombination and replicative transposition [39,40]. Multiple copies of IS*26* with different orientations located on the F18:A-:B1 plasmid are very likely to form cointegrates to promote the transmission of ARGs.

## 5. Conclusions

We reported the high prevalence of *bla*_CTX-M-55_-carrying *E. coli* isolated from duck–fish polyculture farms. Both horizontal transfer and clonal spread contributed to the dissemination of the *bla*_CTX-M-55_ gene among *E. coli* strains isolated from ducks and their environment, and the F18:A-:B1 plasmid may play an important role in the spread of *bla*_CTX-M-55_. Coexistence of *bla*_CTX-M-55_ and other resistance genes (eg., *tetA*, *floR*, *fosA3*, *bla*_TEM_, *aadA5 CmlA*, *InuF*) on the same F18: A-: B1 plasmid may result in the co-selection of these resistance determinants and accelerate the dissemination of *bla*_CTX-M-55_ in *E. coli*. In addition, our study is the first to report the emergence of a *fosA7.5* and *bla*_CTX-M-55_ co-carrying ST602 clone in *E. coli* isolated from ducks and soil. These findings emphasize the importance of the ongoing surveillance of *bla*_CTX-M-55_-positive *E. coli* in duck–fish polyculture farms in China.

## Figures and Tables

**Figure 1 antibiotics-12-00961-f001:**
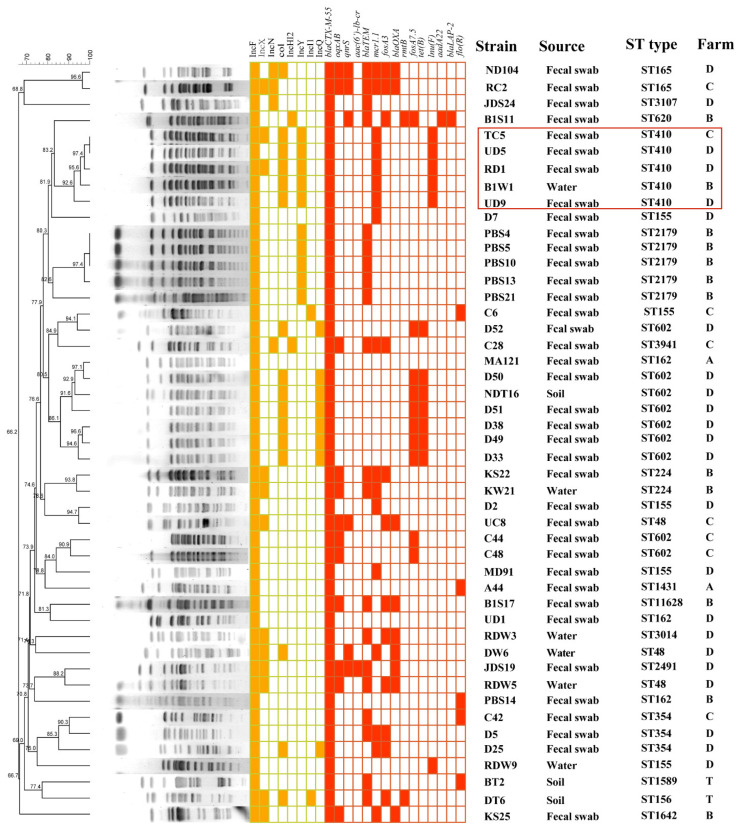
Pulsed-field gel electrophoresis fingerprinting patterns of *XbaI*-digested total DNA preparations from 47 strains harboring *bla*_CTX-M-55_. In yellow is indicated the PBRT classification, in red are indicated antimicrobial resistance genes.

**Figure 2 antibiotics-12-00961-f002:**
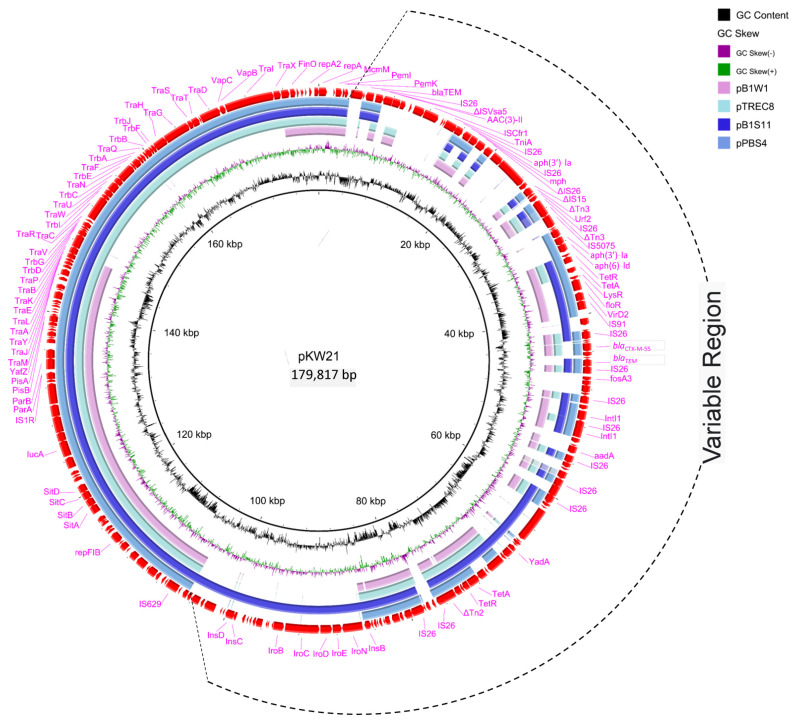
Circular sequence alignments of the plasmids pKW21, pB1W1, pB1S11, pPBS4 and pTREC8. Genes depicted in the outer circle belong to plasmid pKW21, which was included as a reference, and the image was generated using BRIG.

**Figure 3 antibiotics-12-00961-f003:**
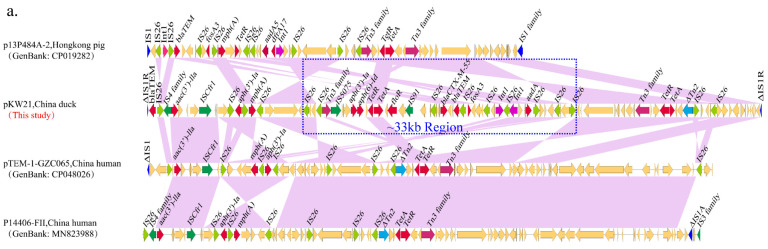
Comparison of the multidrug resistance regions of F18:A-:B1 plasmids identified in this study. (**a**) Comparison of pKW21 with similar plasmids p14406-FII (GenBank accession no. MN823988), p13P484A-2 (GenBank accession no. CP019282) and pTEM-1-GZC065 (GenBank accession no. CP048026). (**b**) Comparison of pB1S11, pPBS4 and pB1W1 with the similar plasmid pGSH8M-2-1 (GenBank accession no. AP019676.1). Genes are denoted by arrows. Genes, mobile elements and other features are colored based on functional classification. Shaded regions denote shared DNA homology (>95% nucleotide identity).

## Data Availability

The whole genome sequence generated in this study is available from the National Center for Biotechnology Information (Accession No. PRJNA934699).

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
