# Peer review of "F18:A-:B1 Plasmids Carrying blaCTX-M-55 Are Prevalent among Escherichia coli Isolated from Duck–Fish Polyculture Farms"

_antibiotics, 2023, doi:10.3390/antibiotics12060961_

Round 1

Reviewer 1 Report

The authors presented a manuscript entitled "F18:A-:B1 plasmids carrying blaCTX-M-55 are prevalent among Escherichia coli isolated from duck-fish polyculture farms" which is a very well-written document with interest for the AMR area and highlighting the importance of the horizontal transfer of the resistance determinants.

My observations are minimal and are listed below:

-Line 36, (E. coli) is not necessary, please erase it.

-Fig 1. Please erase "PFGE-XbaI" legend over the dendrogram. Additionally, the caption must be more informative e.g. In yellow are indicated the PBRT classification.

- Fig 3.  In the third plasmid, there is a truncated class I Integron but the relevance of this finding is not mentioned. The integrons have relevance in AMR and some previous works indicated that IS presence favors the expression of the genes downstream. Maybe you should explore these ideas

-L312, there is a missing period after [13] reference.

- L236, in section 3.4 is not mentioned how the southern blotting was conducted instead is indicated in the section title.

Author Response

Reviewer # 1:

  1. -Line 36, (E. coli) is not necessary, please erase it.

Response: Thank you for your detailed review. We have erased (E. coli) in line 35.

  1. -Fig 1. Please erase "PFGE-XbaI" legend over the dendrogram. Additionally, the caption must be more informative e.g. In yellow are indicated the PBRT classification.

Response: Thank you for your kind comments and advice. We have erased "PFGE-XbaI" legend over the dendrogram of Fig 1 and add the information “In yellow are indicated the PBRT classification, in red are indicated antimicrobial resistance genes” in the caption of Figure 1 in lines 129-130.

  1. - Fig 3. In the third plasmid, there is a truncated class I Integron but the relevance of this finding is not mentioned. The integrons have relevance in AMR and some previous works indicated that IS presence favors the expression of the genes downstream. Maybe you should explore these ideas.

Response: Thank you for very much for your valuable comments and advice. We have added the description of the truncated class I integron in lines 223-224 and marked this integron in Figure 3b. Co-localization of class 1 integrons and ISs with ARGs in Gram-negative bacteria promoting the dissemination of ARGs is already well documented, similar views are common, so we did not emphasize in this study.

References:

Partridge SR. Analysis of antibiotic resistance regions in Gram-negative bacteria. FEMS Microbiol Rev. 2011 Sep;35(5):820-55.

Partridge SR, Kwong SM, Firth N, Jensen SO. Mobile Genetic Elements Associated with Antimicrobial Resistance. Clin Microbiol Rev. 2018 Aug 1;31(4):e00088-17.

Kubomura A, Sekizuka T, Onozuka D, Murakami K, Kimura H, Sakaguchi M, Oishi K, Hirai S, Kuroda M, Okabe N. Truncated Class 1 Integron Gene Cassette Arrays Contribute to Antimicrobial Resistance of Diarrheagenic Escherichia coli. Biomed Res Int. 2020 Jan 31; 2020:4908189.

Sabbagh P, Rajabnia M, Maali A, Ferdosi-Shahandashti E. Integron and its role in antimicrobial resistance: A literature review on some bacterial pathogens. Iran J Basic Med Sci. 2021 Feb;24(2):136-142.

  1. -L312, there is a missing period after [13] reference.

Response: Thank you for your suggestion. We apologize for our carelessness, we have added a period after [13] reference in line 319.

  1. - L236, in section 3.4 is not mentioned how the southern blotting was conducted instead is indicated in the section title.

Response: We are extremely grateful to the reviewer for pointing out this problem. We added a reference to in lines 282-284 to description how the southern blotting was conducted.

Reviewer 2 Report

The present work aims to describe the epidemiological situation of blaCTX-M-55 positive E. coli in duck-fish polyculture farms.

The title is well suited for the paper and the paper itself analyses a scientific topic which has not been extensively explored in previous literature. The One-Health approach used in this study, analyzing not only animal-related but also environmental samples, paired with top notch molecular assays for sample characterization elevates the scientific soundness of the work. 

Below are pointed out two small corrections that I think should be addressed.

line 140-143: you probably mispelled the lincomycin resistance gene, it should be InuF and not IunF. Please make sure to double check the name of the gene

line 240: please state what platform you used to perform 16S rRNA sequencing.

Author Response

Reviewer # 2:

  1. line 140-143: you probably mispelled the lincomycin resistance gene, it should be InuF and not IunF. Please make sure to double check the name of the gene

Response: Thank you for your careful review. We admit our mistakes for our carelessness. We have corrected changing " IunF" to " InuF" and the whole text was carefully checked to avoid the same mistakes.

  1. line 240: please state what platform you used to perform 16S rRNA sequencing.

Response: We deeply appreciate for your suggestion. The 16S rRNA were amplified by the primer reported in “Universal amplification and analysis of pathogen 16S rDNA for classification and identification of mycoplasma like organisms”, and the PCR products were sequenced by Sanger sequencing. We have added this information in line 246.

Reference:

Lee I M . Universal amplification and analysis of pathogen 16S rDNA for classification and identification of mycoplasmalike organisms.[J]. Phytopathology, 1993, 83(8):834-842.

Reviewer 3 Report

The manuscript entitled “F18:A-:B1 plasmids carrying blaCTX-M-55 are prevalent among Escherichia coli isolated from duck-fish polyculture farms” is interesting, relevant, and well written and should be included in the special issue “Molecular Characterization of Gram-Negative Bacteria: Antimicrobial Resistance, Virulence and Epidemiology”.

The study was well conducted, the results are adequately treated, and are presented and discussed in a clear, logical, and coherent way.

Therefore, the manuscript should be considered for publication after some minor corrections/revisions are made.

-

Author Response

Reviewer # 3:

  1. (lines 20-21) – “ … were selected for further study the molecular characteristics and transmission mechanism … ” should be changed to " … were selected to further study the molecular characteristics and transmission mechanism … " or “… were selected forurther study of the molecular characteristics and the transmission mechanism …”

Response: Thank you for your suggestions. We have changed “for” to “to” in line 20.

  1. (line 22) – the abbreviation “WGS” is used but the related full name is not given beforeit. Please add the full name ("whole-genome sequencing (WGS)").

Response: Thank you for your suggestions. We apologize for our carelessness, the first occurrence of abbreviations needs to follow a certain format, we have therefore changed “WGS” to “whole-genome sequencing (WGS)” in line 22.

  1. (lines 44-45) - “… amino acid homology; among these groups, …” ought to be replaced by “… amino acid homology. Among these groups, ….”

Response: Thank you for your careful review. We have changed “…amino acid homology; among these groups…” to “…amino acid homology. Among these groups…” in lines 43-44.

  1. (lines 89 and 93) – abbreviations are again used (“PBRT” and “RST”) without giving the full name first; please correct: PCR-base replicon typing (PBRT) and replicon sequencing typing (RST)

Response: We are extremely grateful to the reviewer for pointing out this problem. We apologize for the similar errors, we have changed “PBRT” and “RST” to “PCR-base replicon typing (PBRT) and replicon sequencing typing (RST)” in line 93 and line 98.

  1. (line 127) – remove the word “were” line 128

Response: Thank you for your advice. We have removed the word “were” in line 132.

  1. (line 128) - replace “… distinct STs.” by “… distinct strains (STs).”

Response: Thank you for your kind suggestion. We have replaced “… distinct STs” by “… distinct strains (STs)” in line 134.

  1. (lines 130-131) – it is suggested to rewrite the sentence " Of the remaining eight ST types, only one strain belonged to each." in order to make it clearer.

Response: Thank you for your careful review. We have rewritten this sentence as “Of the remaining eight ST types, only one strain belonging to each ST type” in line 137. The description will hopefully to make it clearer.

  1. (line 176) - replace “Liner comparison …” by “Linear comparison …”

Response: Thank you for your suggestions. We apologize for our carelessness, the “Liner” has been changed to “Linear” in line 181.

  1. (line 182) - alter “BLAST analysis…” to ” Basic Local Alignment Search Tool (BLAST) analysis…”

Response: We are extremely grateful to the reviewer for pointing out this problem. We apologize for the similar errors again, the“BLAST analysis…”has been changed to “Basic Local Alignment Search Tool (BLAST) analysis…” in line 187.

  1. (lines 187-188) – remove the paragraph between these two lines so that the sentence“Although the F18:A-:B1 plasmid possessed a conserved backbone region, the variable regions …”

Response: Thank you very much for your kind suggestions. We have removed the paragraph between these two lines and changed “Regions” to “region” in line 193.

  1. (line 247) – change the capital letter from "Fosfomycin" to lower case

Response: Thank you for your advice. We apologize for our carelessness, the “Fosfomycin” has been changed to “fosfomycin” in line 253.

  1. (line 269) – the concentrations shown for cefotaxime are shown with only one significantfigure (e.g. 1 mg/L cefotaxime – line 238 and 269; 0.5 mg/L cefotaxime – line 272), so it is questionable why the value for streptomycin appears with 4 significant figures (“2,000 mg/L”)

Response: We are extremely grateful to the reviewer for pointing out this problem. We have read this paragraph carefully and repeatedly, changed “2,000 mg/L of streptomycin” and “200 mg/L sodium azide” to “2 g/L of streptomycin” and “0.2 g/L sodium azide” in line 275 and line 279 respectively.

  1. (line 271) – please add a space after the number 0.22 (i.e., it should be 0.22 μm)

Response: Thank you for your advice. We have added a space after the number 0.22 in line 277.

  1. (line 276) – the subheading should move to the next page (line 279).

Response: Thank you for very much for your advice. The subheading “3.5. DNA extraction and whole-genome sequencing” have been moved to the next page (line 283).

  1. (line 308) – change “ARB, …”at the beginning of the line to “antibiotic resistant bacteria(ARB), …”

Response: Thank you for your careful review. We admit our mistakes for our carelessness. This error occurs frequently in the manuscript, we have checked the first occurrence of all abbreviations and corrected all similar errors. The “antibiotic resistant bacteria” was first described in the line 61, we have changed “antibiotic-resistant bacteria” to “antibiotic-resistant bacteria (ARB)” in lines 61-62, thus “ARB” can be used in line 314.

  1. (line 311) – alter “… MGEs to ….” by “ … mobile genetic elements (MGEs) to …”

Response: Thank you for your careful review. We apologize for the similar errors again, it is the same as your questions 2, 4, 9 and 15 above, we have changed the “MGEs” to “mobile genetic elements (MGEs)” in lines 318-319.

  1. (line 312) - a full stop is missing after "[13]"

Response: Thank you for your careful review. We have added a full stop after “[13]” in line 321.

  1. (line 344) – “MLST results” ought to be changed to “Multilocus sequence typing”

Response: Thank you for your careful review. We are so sorry for the similar errors occurs frequently in this manuscript, we have changed “MLST” to “Multilocus sequence typing” in line 352.

  1. (line 357) – change the semicolon after the word "America" to a comma

Response: Thank you very much for your advice. We have changed the semicolon after the word “America” to a comma in line 365.

  1. (line 375) - change the comma after the word "strains" to a period

Response: We are grateful for the advice. We have changed the comma after the word “strains” to a period in line 383.

  1. References: this section needs to be carefully reviewed and corrected; there are several changes to be made and information to be entered (for example in references 4, 7, 9,10, 11, 12, 17, 19, 26, 27, 32, 33, 36, etc.)

Response: Thank you for your careful review. We have rechecked the reference format of the full text and carefully read the requirements on the official website of the journal, in addition to reviewing the reference format of articles published in previous years. As a result, we reformatted the references according to the journal's submission requirements and we used endnote for literature insertion, but a large number of errors were found in the follow-up check. Therefore, we manually checked and corrected all errors, such as adding missing years, modifying species and genes to italics, etc. We hope that the revised reference format will meet the journal's requirements.

Reviewer 4 Report

The manuscript “F18:A-:B1 plasmids carrying blaCTX-M-55 are prevalent among Escherichia coli isolated from duck-fish polyculture farms” is good piece of work, however there are some minor concerns.

Certain points are reiterated multiple times throughout the introduction, leading to redundancy. For example, the prevalence and transmission of blaCTX-M-55 are mentioned several times. Streamlining the content and avoiding unnecessary repetition would enhance the effectiveness of the introduction. I think overall ESBL producing bacteria are threat to public health.

The study mentions that current data on the prevalence, genetic information, and transmission mechanism of CTX-M-55-producing E. coli from duck-fish polyculture farms are limited. However, it would be helpful to elaborate on why this research gap exists and how filling this gap by this study would advance the field.

The quality of English language is good enough. 

Author Response

Response to review 4's comments:

  1. Certain points are reiterated multiple times throughout the introduction, leading to redundancy. For example, the prevalence and transmission of blaCTX-M-55 are mentioned several times. Streamlining the content and avoiding unnecessary repetition would enhance the effectiveness of the introduction. I think overall ESBL producing bacteria are threat to public health.

Response: We are extremely grateful to the reviewer for pointing out this problem. We have read the abstract and introduction carefully and repeatedly, we deleted “the molecular characteristics and transmission mechanism of blaCTX-M-55 in E. coli” of abstract in line 21 to avoid redundancy.

  1. The study mentions that current data on the prevalence, genetic information, and transmission mechanism of CTX-M-55-producing coli from duck-fish polyculture farms are limited. However, it would be helpful to elaborate on why this research gap exists and how filling this gap by this study would advance the field.

Response: We deeply appreciate the reviewer’s suggestion. We added “However, CTX-M-55 has been widely reported in E. coli isolated from food animals, pets and humans in China” and “Our findings emphasize the importance of the surveillance of ESBL producing E. coli in duck-fish polyculture farms in China and provide knowledge for further One Health studies to control the spread of resistant bacteria from food animals to humans” in the in the last paragraph of introduction in lines 66-74. The added description will hopefully to explain why this research gap exists and how filling this gap by this study would advance the field.
